# Everolimus Acts in Synergy with Vinorelbine to Suppress the Growth of Hepatocellular Carcinoma

**DOI:** 10.3390/ijms25010017

**Published:** 2023-12-19

**Authors:** Hung Huynh, Wai Har Ng, Khee Chee Soo

**Affiliations:** 1Laboratory of Molecular Endocrinology, National Cancer Centre Singapore, Singapore 168583, Singapore; nmsnwh@nccs.com.sg; 2Division of Surgery and Surgical Oncology, National Cancer Centre Singapore, Singapore 168583, Singapore

**Keywords:** hepatocellular carcinoma, mTOR inhibitor, microtubule inhibitor, vessel normalization, hypoxia

## Abstract

Hepatocellular carcinoma (HCC) is a challenging cancer to treat, as traditional chemotherapies have shown limited effectiveness. The mammalian target of rapamycin/sirolimus (mTOR) and microtubules are prominent druggable targets for HCC. In this study, we demonstrated that co-targeting mTOR using mTOR inhibitors (everolimus and sirolimus) along with the microtubule inhibitor vinorelbine yielded results superior to those of the monotherapies in HCC PDX models. Our research showed that the vinorelbine arrests cells at the mitotic phase, induces apoptosis, and normalizes tumor blood vessels but upregulates survivin and activates the mTOR/p70S6K/4EBP1 pathway. The addition of the everolimus significantly improved the tumor response to the vinorelbine, leading to improved overall survival (OS) in most tested orthotopic HCC PDX models. The mechanistic investigation revealed that this marked antitumor effect was accompanied by the downregulations of mTOR targets (p-p70S6K, p-4EBP1, and p-S6K); several key cell-cycle regulators; and the antiapoptotic protein survivin. These effects did not compromise the normalization of the blood vessels observed in response to the vinorelbine in the vinorelbine-sensitive PDX models or to the everolimus in the everolimus-sensitive PDX models. The combination of the everolimus and vinorelbine (everolimus/vinorelbine) also promoted apoptosis with minimal toxicity. Given the cost-effectiveness and established effectiveness of everolimus, and especially sirolimus, this strategy warrants further investigation in early-phase clinical trials.

## 1. Introduction

Hepatocellular carcinoma (HCC) is the second most common cause of cancer-related deaths worldwide [1,2]. More than 80% of HCCs arise in liver cirrhosis from various etiologies, including hepatitis B and hepatitis C infections, excessive alcohol consumption, diabetes mellitus, and non-alcoholic fatty liver disease (NAFLD) [3]. The presence of intratumor heterogeneity in HCC makes the clinical management of the disease challenging. Surgical resection and liver transplantation offer a potential cure for patients with early-stage HCC [4]. Unfortunately, many patients diagnosed with HCC are not amenable to curative resection. Systemic agents, such as sorafenib and lenvatinib, have demonstrated a survival benefit for patients with advanced HCC [5,6,7]. Despite the benefits of sorafenib [5,7] and lenvatinib [6] monotreatments, their impacts are modest and are often accompanied by the development of drug resistance. Recently, systemic therapies, such as atezolizumab plus bevacizumab and durvalumab plus tremelimumab, have been considered as preferable to first-line therapy options, except for patients with high-risk stigmata of variceal or other gastrointestinal bleeding and those with liver cirrhosis in whom immune-based regimens are contraindicated (e.g., owing to severe autoimmune disorders or liver transplantation) [8,9]. Investigations into the combination of lenvatinib plus anti-PD1 in unresectable HCC have also been conducted [10]. Two other multikinase inhibitors, regorafenib [11] and cabozantinib [12], have been approved as second-line therapies after showing a significantly improved OS versus a placebo in HCC patients. Given the aggressive nature of HCC and its propensity for early dissemination inside the liver [13], there is an urgent need for novel, effective, and affordable treatments for this lethal disease.

The aberrant activation of the mammalian target of rapamycin/sirolimus (mTOR) is common in HCC (∼45% of cases), and its overexpression is associated with a poor prognosis in HCC patients [14,15]. mTOR is a key effector in the PI3K/AKT/mTOR pathway and plays a critical role in regulating cell proliferation and survival. The downstream targets of mTOR include ribosomal p70S6K kinase and the eukaryotic initiation factor eIF4E binding protein (4E-BP1), which regulate the cell cycle, growth, and protein synthesis. Recently, it has been shown that the mTOR inhibitor everolimus specifically inhibits DNA synthesis in hepatocytes and markedly delays liver tumor development induced by DNA damage, indicating that mTOR activation has a substantial effect on hepatocarcinogenesis [16]. Therefore, mTOR is a prominent target for HCC therapy. A global phase III study of everolimus in patients with HCC after progression on sorafenib (everolimus for liver cancer evaluation-1 (EVOLVE-1)) failed to reach its primary endpoint in improving OS with everolimus [17]. Because the effect of mTOR targeting is largely cytostatic [18], the utility of mTOR inhibitors as single agents for cancer treatment may be limited. It is anticipated that the combination of an mTOR inhibitor with a cytotoxic agent may represent a more effective therapeutic regimen than mTOR targeting alone.

Microtubule-related cellular assembly and organization play a crucial role in HCC development [19,20,21] by affecting not only mitosis but also the cytoskeletal shape, cell motility, intracellular protein, and organelle transport [22]. Thus, microtubule-targeting agents inhibit cell proliferation by inducing cell-cycle arrest and apoptosis, making them a potentially important therapeutic target for HCC [21]. Among these microtubule-targeting agents, vinorelbine has demonstrated radio-enhancer activity, even at low doses [23,24]. In trials showing safety and clinical efficacy, it has been used as a standard concurrent chemo-radiotherapy regimen for lung cancer patients [25,26]. Our previous studies have shown that when given in combination with the FGFR1-3 inhibitor infigratinib [27], the FGFR4 inhibitor FGF401 [28], or radiotherapy [29], vinorelbine was effective in inhibiting HCC growth in vivo.

Both microtubules and the mTOR pathway are considered as druggable targets, and therapeutic compounds targeting these have already been developed and are readily available for other indications. In the present study, we sought to determine the antitumor activity of vinorelbine in combination with mTOR inhibitors (everolimus and sirolimus) and to understand the mechanisms underlying the inhibition of the growth of HCC tumors by vinorelbine in combination with mTOR inhibitors. First, we assessed whether the addition of everolimus improved the antitumor activity of vinorelbine. Second, we investigated whether vinorelbine affected tumor angiogenesis, tumor hypoxia, and the OS of mice bearing orthotopic HCC tumors. Finally, we determined whether the slowing of the tumor growth resulted from reduced tumor cell proliferation, increased tumor cell death, or both. Herein, we report the effects of vinorelbine monotherapy alone or in combination with either everolimus or sirolimus in patient-derived xenograft (PDX) models of human HCC.

## 2. Results

### 2.1. Vinorelbine Demonstrated Antimitotic and Apoptotic Activities, Reduced Tumor Hypoxia via Blood-Vessel Normalization, and Upregulated Downstream Targets (p-p70S6K/4EBP1/Survivin) of the mTOR Pathway in HCC PDX Models

First, we examined the antitumor effects of a dose of 3 mg/kg of vinorelbine administered twice per week. As shown in Figure 1 and Appendix A, compared with the vehicle control, vinorelbine significantly decreased tumor growth rates in the HCC13–0109, HCC25–0705A, HCC13–0212, HCC19–0913, HCC05–0614, HCC19–0509, HCC26–0808B, and HCC24–0309 PDX models but showed only a modest effect on the HCC29–0909A, HCC27–1014, HCC01–0708, HCC09–0913, and HCC06–1009 PDX models. Except for in the HCC05–0411B PDX model, the vinorelbine-treated tumors were significantly smaller than those in the vehicle group (Figure 1 and Appendix A). Out of the 15 HCC PDX models treated with vinorelbine, five HCC PDX models (33.3%, i.e., HCC13–0109, HCC13–0212, HCC19–0509, HCC19–0913, and HCC25–0705A) showed T/C ratios of <0.42 (Table 1).

Compared with the vehicle treatment, vinorelbine had significant effects on the HCC19–0913, HCC13–0109, HCC25–0705A, HCC13–0212, and HCC24–0309 PDX models, resulting in increases in the numbers of p-histone H3 Ser10-positive cells, blood vessels, and cleaved PARP-positive cells. In the vehicle-treated tumors, the blood vessels displayed irregular shapes and tortuosity, which were indicative of vascular remodeling. Conversely, the vinorelbine-treated tumors exhibited slim, capillary-like vessels, as depicted in Figure 2 and Appendix A. However, in the HCC29–0909A, HCC09–0913, HCC30–0805B, and HCC29–1104 PDX models treated with vinorelbine, there were only modest increases in the numbers of apoptotic cells and mitotic cells (Appendix A). As shown in Table 1, these models had a T/C ratio of >0.42, suggesting their relative resistance to vinorelbine. Interestingly, vinorelbine did not significantly induce the blood-vessel phenotype in the HCC29–0909A, HCC09–0913, HCC30–0805B, and HCC29–1104 PDX models, as shown in Appendix A. Our data highlight a correlation between the impact of vinorelbine on blood vessels and its observed efficacy. Similar results were observed when analyzing other HCC PDX models treated with the vehicle or vinorelbine.

In the vinorelbine-sensitive HCC13–0212 model (as shown in Appendix A), treatment with 3 mg/kg of vinorelbine in mice bearing tumors led to a significant increase in the percentage of p-histone H3 Ser10-positive cells within 24 h. The maximum level was reached by 48 h, and the level returned to the baseline by 72 h after the treatment. Notably, significant apoptosis, as indicated by the number of cleaved PARP-positive cells, was observed within 24 h and reached its maximum at 72 h following the vinorelbine treatment. Moreover, the microvessel density significantly increased and reached a peak at 72 h.

To assess the functionality of the increased network of vessels in the vinorelbine-treated tumors, we intravenously administered biotinylated tomato lectin (from *Lycopersicon esculentum*) to the vehicle- and vinorelbine-treated mice. This enabled us to label the murine vascular endothelium and examine the perfused vasculature. Subsequently, we infused pimonidazole HCl to measure the hypoxic microenvironment within the tumors. As depicted in Figure 2 and Appendix A, fewer lectins were bound to the blood vessels of the vehicle-treated tumors in the HCC19–0913, HCC13–0109, HCC25–0705A, HCC13–0212, and HCC24–0309 PDX models. Many regions showed no lectin immunostaining, suggesting that a high proportion of blood vessels in the vehicle-treated tumors were nonfunctional. Large sections of the tumors exhibited positive staining with Hypoxyprobe, indicating the presence of hypoxic regions. In contrast, most of the capillary-like blood vessels induced by the vinorelbine treatment stained positively for biotinylated lectin, implying that they were well perfused. Additionally, Hypoxyprobe staining was negative across large sections of the tumors, indicating that these regions were well oxygenated and that vinorelbine restored intratumoral oxygenation. Taken together, the analyses of tumor hypoxia and perfusion suggest that vinorelbine normalizes blood-vessel functions.

A western blot analysis revealed that vinorelbine treatment for 3 days caused significant increases in the levels of p-p70S6K (Thr389 and Thr421/424) and p-4EBP1 (Thr70) in vinorelbine-sensitive HCC19–0913 tumors, suggesting that the downstream targets of mTOR were activated (Appendix A).

We next investigated the antitumor effects of everolimus monotherapy. As shown in Figure 1, Table 1, and Appendix A, the oral administration of 2 mg/kg of everolimus (or sirolimus) effectively inhibited the growth of the HCC24–0309, HCC25–0705A, HCC05–0411B, HCC19–0509, and HCC26–0808B PDX models, indicating that these models were relatively sensitive to mTOR inhibitors. The tumor sizes were approximately three- to eight-fold smaller in the everolimus-treated group than in the vehicle-treated groups. In contrast, the tumor volumes of the everolimus-treated HCC01–0708, HCC05–0614, HCC09–0913, HCC13–0109, HCC19–0913, HCC27–1014, and HCC29–0909A PDX models exhibited either insignificant reductions or only modest reductions in comparison with those of the vehicle-treated PDX models, suggesting that these models were relatively resistant to everolimus. The everolimus treatment was considered as being effective (T/C < 0.42) in 7 out of the 14 (50%) HCC PDX models (i.e., HCC05–0411B, HCC06–1009, HCC13–0212, HCC19–0509, HCC24–0309, HCC25–0705A, and HCC26–0808B). It also significantly increased the numbers of both total blood vessels and productive blood vessels while mitigating tumor hypoxia in the mTOR-sensitive HCC25–0705A and HCC24–0309 PDX models (Appendix A) but not in the vinorelbine-sensitive HCC19–0913, HCC13–0109, and HCC13–0212 PDX models (Figure 2 and Appendix A). The everolimus-treated HCC13–0109 and HCC13–0212 tumors had fewer total blood vessels and productive blood vessels than the vehicle-treated tumors did (Appendix A, respectively), indicating that everolimus possessed antiangiogenic properties in these models.

### 2.2. Vinorelbine Acted in Synergy with Everolimus (or Sirolimus) to Inhibit Tumor Growth

Vinorelbine induced blood-vessel normalization (Figure 2 and Appendix A) and reactivated the downstream targets of mTOR (Appendix A). These observations prompted us to investigate the potential antitumor effects of vinorelbine in combination with mTOR inhibitors (everolimus and sirolimus) in the HCC PDX models, with each exhibiting varying levels of sensitivity to either vinorelbine or mTOR inhibitors. Our hypothesis was that the normalization of blood vessels induced by either vinorelbine or everolimus would enhance drug delivery to tumors, resulting in greater inhibition of tumor growth and improved survival in mice bearing HCC tumors.

To investigate this hypothesis, a total of 15 HCC PDX models were given the following treatments: (a) vehicle, (b) vinorelbine, (c) everolimus, or (d) everolimus/vinorelbine, as described in Section 4. The results showed that within this cohort of 15 HCC PDX models, four exhibited sensitivities to the mTOR inhibitor, two were sensitive to vinorelbine, three were sensitive to both vinorelbine and the mTOR inhibitor, and six were resistant to both vinorelbine and the mTOR inhibitor (Table 1).

Among these HCC PDX models, 14 (excluding the HCC26–0808B model, which was too sensitive to everolimus) exhibited a greater inhibition in tumor growth and a greater reduction in tumor burden with everolimus/vinorelbine than with everolimus or vinorelbine alone (Figure 1 and Appendix A). Similar results were obtained when everolimus was substituted with sirolimus for the HCC06–1009, HCC13–0212, HCC19–0913, and HCC30–0805B PDX models (Table 1 and Appendix A). It is noteworthy that all 14 HCC PDX models (100%) treated with everolimus/vinorelbine, including those that were relatively resistant to both everolimus and vinorelbine monotherapies (HCC01–0708, HCC05–0614, HCC09–0913, HCC27–1014, and HCC29–0909A), exhibited T/C ratios of <0.37, surpassing the 0.42 threshold set by the Cancer Therapy Evaluation Program (CTEP) of the Investigational Drug Branch (IDB, Bethesda, MD, USA) at the National Cancer Institute [30] (Table 1). These results indicate that the addition of an mTOR inhibitor significantly enhanced the antitumor activity of vinorelbine in the HCC PDX models. Furthermore, no significant loss of bodyweight (*p* = 0.6543) or other clinical signs of toxicity were observed in the mice within the treatment groups when compared with the vehicle group.

To validate these findings, we conducted an analysis of the liver enzymes in the sera prepared from mice bearing HCC PDX tumors treated with the vehicle, everolimus, vinorelbine, and everolimus/vinorelbine. As shown in Table 2, the daily treatment of the mice with everolimus resulted in modest elevations in ALT, ALP, AST, and TBIL. This observation aligns with the safety profiles of everolimus in human studies [17,31,32], where enzyme elevations occur in up to a quarter of the patients taking everolimus, but the abnormalities are usually mild and rarely require dose modification or discontinuation. In comparison with everolimus, vinorelbine caused greater elevations in ALT, ALP, AST, and TBIL, suggesting mild liver dysfunction. Vinorelbine, but not everolimus, caused a mild elevation in BUN (1.2-fold). In clinics, vinorelbine treatment is associated with elevations in serum aminotransferase levels in from 5% to 10% of patients [33]. Despite being cytotoxic for cancer cells and metabolized actively by the liver, vinorelbine has only rarely been associated with significant hepatic toxicity [33]. There were further elevations in BUN, ALT, ALP, AST, and TBIL when everolimus was combined with vinorelbine, but the elevations were not significant in comparison with those of vinorelbine monotherapy (Table 2). No significant changes in the levels of serum glucose (GLU) and albumin (ALB) within the treatment groups were observed when compared with those in the vehicle group. These data suggest that vinorelbine and everolimus/vinorelbine caused mild hepatic toxicity.

We then assessed the effects of the treatments on the tumor blood vessels, blood-vessel normalization, tumor hypoxia, tumor proliferation, and apoptosis. When compared with tumors treated with either the vehicle or everolimus monotherapy, the tumors treated with vinorelbine or everolimus/vinorelbine—particularly in the HCC19–0913, HCC13–0109, and HCC13–0212 PDX models—exhibited a two- to five-fold increase in the number of p-histone H3 Ser10-positive cells (Figure 2 and Appendix A). However, when compared with the treatment with the vinorelbine monotherapy, the addition of everolimus had varying effects, leading to either no significant changes (in HCC19–0913, HCC13–0212, and HCC13–0109; Figure 2 and Appendix A) or a significant reduction (in HCC25–0705A; Appendix A) in the percentage of p-histone H3 Ser10-positive cells.

In the vinorelbine-sensitive HCC PDX models (HCC19–0913 and HCC13–0109), the vinorelbine/mTOR-sensitive model (HCC13–0212), and the mTOR-sensitive model (HCC24–0309), everolimus/vinorelbine displayed from 1.5- to 10-fold increases in the percentage of cleaved PARP-positive cells, as well as significantly higher (from 1.5 to 2.5 fold) total blood-vessel counts compared with those of the vinorelbine and everolimus monotherapies (Figure 2 and Appendix A). Additionally, when we performed lectin perfusion and hypoxia immunostaining, we observed that the capillary-like blood vessels in these models were well-perfused and functional, effectively mitigating tumor hypoxia. However, everolimus/vinorelbine did not cause any significant changes in the normalization of blood vessels or tumor hypoxia, which were observed in response to vinorelbine in vinorelbine-sensitive PDX models or to everolimus in the everolimus-sensitive PDX models that were tested (Figure 2 and Appendix A).

In the HCC25–0705A (mTOR inhibitor/vinorelbine-sensitive) model, the addition of everolimus to vinorelbine did not result in any further increases in cleaved PARP positivity in comparison with those of the vinorelbine or everolimus monotherapies (Appendix A, respectively). Similarly, for the HCC PDX models that were resistant to both vinorelbine and mTOR inhibitors (HCC29–0909A, HCC09–0913, HCC30–0805B, and HCC29–1104), everolimus/vinorelbine did not lead to significant changes in the numbers of p-histone H3 Ser10-positive cells, cleaved PARPs, or total blood vessels in comparison with those of the treatments with the vehicle, everolimus, or vinorelbine alone (Appendix A). In summary, these observations suggest that the combination of vinorelbine with mTOR inhibitors effectively slowed tumor growth by promoting apoptosis, enhancing vascular function, and decreasing tumor hypoxia. To validate these findings, we conducted a propidium iodide (PI) flow cytometry analysis to determine the percentage of apoptotic cells and the frequency of mitotic cells 24 h after the treatments. As shown in Table 3, the everolimus treatment caused a significant elevation in the percentage of G1 cells but a decrease in the percentage of G2/M cells. In contrast, the number of cells in the G2/M phase in vinorelbine was elevated, while the percentage of G1 cells was reduced. The percentage of G2/M-phase cells in the combination treatment was higher than that in the vinorelbine treatment alone. Additionally, vinorelbine and everolimus/vinorelbine significantly increased the percentage of sub-G1 cells, which was consistent with the induction of apoptosis.

### 2.3. Everolimus Potentiated the Antitumor Activity of Vinorelbine to Inhibit Tumor Growth, Cell Proliferation, and the Expression of Positive Cell-Cycle Regulators and Prolonged the Survival of Mice Bearing HCC Tumors

To gain a deeper understanding of the mechanism(s) by which vinorelbine/mTOR inhibitors exert their antitumor activity in HCC PDX models, a western blot analysis was conducted. Figure 3 illustrates the results of the western blot analysis conducted on the HCC25–0705A model after treatment with vinorelbine, everolimus, or everolimus/vinorelbine. Compared with the vehicle, the vinorelbine treatment increased the levels of various proteins, including p-p70S6K (Thr389), p-p70S6K (Thr421/Ser424), p-4EBP1 (Thr70), p-Cdc2 (Tyr15), p-Cdk2 (Thr14/Tyr15), p-Rb (Ser807/811), Cyclin B1, p27, survivin, and cleaved caspase 3/7. In contrast, the everolimus treatment led to significant reductions in the levels of p-mTOR (Ser2448), p-p70S6K (Thr421/Ser424), p-Cdc2 (Tyr15), p-Cdk2 (Thr14/Tyr15), p-Rb (Ser807/811), Rb, Cyclin B1, Cyclin D1, Cdc25, p-Cdc25C (Ser216), and survivin. Additionally, the everolimus treatment led to increases in the levels of p-AKT (Ser473) and p27 but undetectable levels of p-4EBP1 (Thr70) and p-S6R (Ser235/236) when compared with those of the vehicle treatment. It is noteworthy that when everolimus was combined with vinorelbine, the combination abolished the vinorelbine-induced upregulations of p-p70S6K (Thr389), p-p70S6K (Thr421/Ser424), p-4EBP1 (Thr70), p-S6R (Ser235/236), p-Cdc2 (Tyr15), p-Cdk2 (Thr14/Tyr15), Cdc25C, p-Rb (Ser807/811), Rb, Cyclin B1, and survivin. However, this combination led to significant increases in the levels of p-AKT (Ser473), p27, cleaved caspase 3, and cleaved caspase 7. Additionally, modest reductions in the levels of p-ERK1/2 and Cyclin D1 were observed in the everolimus/vinorelbine treatment group.

To validate these findings, we conducted western blot analyses targeting the same proteins in various HCC PDX models. The obtained results demonstrated some variations in the effects of everolimus, vinorelbine, and everolimus/vinorelbine on specific proteins in these different models. For example, as shown in Appendix A, when the HCC13–0212 PDX model was treated with vinorelbine, significant increases in the levels of p-mTOR (Ser2448), p-p70S6K (Thr389), p-p70S6K (Thr421/Ser424), p-S6R (Ser235/236), p-Cdc2 (Tyr15), p27, and survivin were observed, but the levels of Cdc25C and p-4EBP1 (Thr70) significantly decreased. Nonetheless, when everolimus was combined with vinorelbine, the combination suppressed the vinorelbine-induced upregulations of p-mTOR (Ser2448), p-p70S6K (Thr421/Ser424), p-p70S6K (Thr389), p-Cdc2 (Tyr15), p-Cdk2 (Thr14/Tyr15), and survivin. In contrast, increases in the levels of everolimus-induced p-AKT (Ser437) and cleaved caspase 7 were observed, and the expressions of Cdc25 and Cdc2 became undetectable. Furthermore, there were modest decreases in the levels of p-ERK1/2 and total Rb in the everolimus/vinorelbine-treated tumors. Similar results were obtained when analyzing the HCC19–0913 PDX model treated with everolimus/vinorelbine (Appendix A). Equivalent results were obtained when sirolimus was substituted for everolimus (Appendix A).

As shown in Figure 4, the Kaplan–Meier survival analysis revealed that all the HCC orthotopic mice treated with the vehicle reached a moribund state on days 48 (HCC13–0212), 58 (HCC24–0309), and 52 (HCC25–0705A). Both the everolimus and vinorelbine monotherapies significantly extended the survival time of the HCC orthotopic mice (*p* < 0.01; log-rank test). In the HCC13–0212 orthotopic model, the vinorelbine-treated group exhibited a longer survival time (92 days) than that of the everolimus-treated group (74 days). Conversely, in the HCC24–0309 and HCC25–0705A orthotopic models, the everolimus-treated mice had longer survival times (102 days and 106 days) than the vinorelbine-treated mice did (66 days and 78 days), respectively. These findings suggest that the efficacy of the everolimus and vinorelbine monotherapies in prolonging the survival of HCC orthotopic models was model-dependent. Furthermore, there were significant differences in survival time between the everolimus- and vinorelbine-treated HCC orthotopic mice (*p* < 0.05; log-rank test). Notably, the group of HCC orthotopic mice treated with everolimus/vinorelbine had the longest survival times of all the treated groups. They survived until days 130, 150, and 136 for the HCC13–0212, HCC24–0309, and HCC25–0705A orthotopic models, respectively (*p* < 0.01; log-rank test). Our data show that everolimus/vinorelbine is superior to the everolimus and vinorelbine monotherapies in improving the OS of mice bearing orthotopic HCC tumors.

## 3. Discussion

HCC ranks as the second most common cause of cancer-related death globally [1]. Given that a significant number of HCC patients present with an advanced and inoperable form of the disease, there remains an unmet need for improved systemic treatment. Although sorafenib [5,7] and lenvatinib [6] have made notable progress in this regard, the development of alternative systemic therapies is essential for patients who either become refractory to or are ineligible for sorafenib or lenvatinib treatment.

It has been reported that mTOR signaling is associated with cancer-cell resistance to microtubule-targeting agents, and the everolimus treatment effectively inhibits the mTOR-associated resistance [35]. We hypothesized that the co-targeting of an mTOR inhibitor and a microtubule inhibitor in HCC might yield better results than those of monotherapies. To test this hypothesis, we investigated the antitumor effectiveness of mTOR-targeting agents (everolimus and sirolimus) in combination with vinorelbine (a common microtubule-destabilizing agent) in HCC PDX models. Additionally, we aimed to gain a better understanding of the molecular mechanisms underlying the antitumor effects of vinorelbine both as a standalone treatment and in combination with everolimus or sirolimus. Herein, we report the effects of vinorelbine, everolimus, and everolimus/vinorelbine or sirolimus/vinorelbine on tumor growth, tumor angiogenesis, cell proliferation, and apoptosis in 15 HCC PDX models. When administered at a dose of 3 mg/kg twice per week, vinorelbine demonstrated a T/C ratio of <0.42 in 4 out of the 15 (26.7%) HCC PDX models that were tested (i.e., HCC13–0212, HCC19–0509, HCC13–0109, and HCC19–0913), meeting the criteria set by the Cancer Therapy Evaluation Program (CTEP) of the Investigational Drug Branch (IDB) at the National Cancer Institute [30], indicating that vinorelbine was efficacious in these four HCC PDX models. The vinorelbine promoted the mitotic arrest and apoptosis of cancer cells, blood-vessel normalization, as well as the inhibition of tumor hypoxia, which was exclusively observed in the vinorelbine-responsive PDX models. The western blot analysis revealed that the vinorelbine reactivated the p70S6K/4EBP1 pathway and caused the upregulations of survivin and positive cell-cycle regulators in vinorelbine-treated tumors.

The mTOR monotherapy using either everolimus or sirolimus effectively inhibited the downstream targets of the mTOR and survivin, resulting in tumor growth inhibition in 7 out of the 15 (46.7%) HCC PDX models (HCC05–0411B, HCC06–1009, HCC13–0212, HCC19–0509, HCC24–0309, HCC25–0705A, and HCC26–0808B). The mTOR inhibitors promoted blood-vessel normalization, leading to a reduction in tumor hypoxia. These observations primarily occurred in the mTOR-sensitive HCC PDX models but not in the mTOR-resistant HCC PDX models, such as HCC29–0909A, HCC09–0913, HCC30–0805B, and HCC29–1104 (Appendix A). Among the 15 PDX HCC models that were tested, p-p706K (Thr421/Ser424) was overexpressed at the baseline in HCC25–0705A (Figure 3), and HCC05–0411B expressed a mutant tuberous sclerosis complex 2 (TSC2) gene [36]. As predicted, the everolimus and sirolimus monotreatments effectively suppressed the activation of the mTOR pathway and reduced the levels of survivin and positive cell-cycle regulators, such as p-Cdc2 (Tyr15), p-Cdk2 (Thr14/Tyr15), p-Rb (Ser807/811), Cyclin B1, Cyclin D1, Cdc25, and p-Cdc25C (Ser216), in HCC25–0705A. Some reports have indicated that the TSC2 status may be related to mTOR pathway activation because TSC2 regulates AKT/p70S6K/4EBP1 [36]. Therefore, the inactivation of TSC2 through mutations or the overexpression of the mTOR pathway leads to the activation of the p70S6K/4EBP1 pathway, contributing to the observed sensitivity of the HCC05–0411B and HCC25–0705A PDX models to mTOR inhibitors [36,37]. Previous studies of other cancers [37,38] and our study [36] have indicated that mTOR targeting may induce cytostatic effects rather than the effective eradication of tumor cells, suggesting the potential advantage of combining mTOR targeting with cytotoxic agents. In the search for a rational combination, the combination of vinorelbine with everolimus was chosen based on the evidence that targeting microtubules with vinorelbine in HCC has recently been shown to be active against HCC [21,27,28,29], and vinorelbine has been used to treat other cancer types [23,24,25,26].

The formation of new blood vessels is an integral part of endothelial cell migration and tumor progression. Two approaches for inhibiting vascular formation are the inhibition of angiogenesis and the disruption of the integrity of the existing tumor vasculature using vascular-disruptive agents [39]. Although vinca alkaloids have been shown to damage the tumor vasculature in animal models [40], we observed that the vinorelbine treatment increased the intratumoral blood-vessel density and blood-vessel normalization in vinorelbine-responsive HCC PDX models. In comparison with the blood vessels in the vehicle-treated tumors, the blood vessels in the vinorelbine-treated tumors exhibited either morphological slimness in the vinorelbine-sensitive PDX models (HCC19–0913, HCC13–0109, HCC13–0212, HCC25–0705A, and HCC24–0309; Figure 2 and Appendix A) or no changes in the vinorelbine-resistant PDX models (HCC29–0909A, HCC09–0913, HCC30–0805B, and HCC29–1104; Appendix A). These observations suggest that vinorelbine induces alterations in the tumor vasculature that are specific and dependent on the degree of sensitivity of each model to the vinorelbine.

The increase in the intratumoral blood-vessel density following the vinorelbine treatment may be attributed to the accumulation of bone-marrow-derived cells (BMDCs) recruited from adjacent tissues. The accumulation of BMDCs in tumors has been demonstrated to stimulate new vessel formation and facilitate vasculature recovery [41,42]. Although the pro-vascular effect of recruited BMDCs has been implicated in tumor protection and disease relapse [42,43], the increases in blood-vessel density and normalization induced by the vinorelbine are associated with apoptosis and antimitotic effects. It remains unclear whether the antitumor activity exhibited by the vinorelbine is a result of classical antimitotic effects on the microtubule dynamics.

The exact mechanisms underlying the ability of the everolimus to induce blood-vessel normalization remain to be elucidated. We found that although the angiogenic vessels in the vasculatures of the vehicle-treated tumors were hyperdilated, distorted, and nonfunctional, most vessels in the vasculatures of the mTOR-sensitive treated tumors were slim, elongated, regularly shaped, and fully functional, as determined through lectin perfusion analysis. It remains to be determined whether the inhibition of the downstream targets of mTOR with everolimus leads to reductions in the vascular endothelial growth factor expression and subsequent blood-vessel normalization. The decrease in tumor hypoxia in these treated tumors suggests that the dense capillary-like network of vessels restores the local oxygen concentration. These phenomena appear to depend on the degree of sensitivity of the tumors to everolimus because they did not occur in the tumors that were relatively resistant to the everolimus or both vinorelbine and everolimus.

The everolimus/vinorelbine treatment consistently exhibited the most favorable effects, even in the HCC01–0708, HCC09–0913, HCC05–0614, HCC27–1014, and HCC29–0909A PDX models (Table 1), which were relatively resistant to both everolimus and vinorelbine (with a T/C ratio of >0.42). Compared with the vehicle and monotherapy groups, the everolimus/vinorelbine-treated groups showed a significant reduction in tumor burden. Previous studies have reported that FXYD-domain-containing ion-transport regulator 5 (FXYD5) enhances the resistance of HCC cells to sorafenib by activating the AKT/mTOR-signaling pathway [44]. Additionally, the downregulation of the argininosuccinate synthase 1 (ASS1) expression by LINC01234 leads to an increased aspartate level and activation of the mammalian target of rapamycin pathway [45]. It is possible that the observed elevation in mTOR signaling in the vinorelbine-treated group resulted from the upregulation of FXYD5 or downregulation of the ASS1 expression. Thus, inhibiting the mTOR-signaling pathway by adding everolimus to vinorelbine can attenuate tumor growth and sensitize vinorelbine-resistant HCC tumors.

The mechanistic investigation revealed that the marked antitumor effect of everolimus/vinorelbine was accompanied by the downregulations of p-p70S6K, p-4EBP1, p-S6K, and survivin, while these factors were upregulated in the vinorelbine monotherapy group. This was further corroborated by the increases in mitotic arrest and apoptosis (Table 3). Our current study showed that everolimus, in combination with vinorelbine, induced no significant changes in vinorelbine-induced or everolimus-induced blood-vessel normalization, leading to a reduction in tumor hypoxia (Figure 2; Appendix A). The everolimus/vinorelbine combination significantly improved the OS of the mice bearing HCC orthotopic tumors, without significant toxicity, as evidenced by the bodyweight, normal food and water intakes, normal social interactions, and activity levels of the mice. Mild liver toxicity was observed, as determined by the mild elevations in sera derived from the vinorelbine and everolimus/vinorelbine treatments (Table 2). Our findings demonstrated that the potent anticancer activity of this co-targeting strategy was indeed mediated in part by blood-vessel normalization and the inhibition of proteins involved in survival and proliferation. Indeed, recent in vivo and in vitro studies have shown that a combination of the mTOR inhibitor temsirolimus and low-dose vinblastine had a marked antitumor effect [43]. Previous studies have demonstrated that this combination also induced a significant reduction in the microvessel density in tumors when compared to those treated with a vehicle control, indicating potential antiangiogenic activity [46,47]. However, we observed no significant changes in the microvessel density, lectin perfusion, or tumor hypoxia when the vinorelbine-sensitive or everolimus-sensitive tumors were treated with either the everolimus/vinorelbine or sirolimus/vinorelbine combination. It is possible that the differences between the results of the present study and the study performed by Qian Zhou et al. [46] could be due to the mode of action of the two inhibitors (stabilizing patupilone versus vinorelbine) and the HCC models employed in the two studies. We studied the effects of a microtubule-destabilizing agent, vinorelbine, using HCC PDX models, while Qian Zhou et al. [46] investigated the effects of a microtubule-stabilizing agent, patupilone, using a Hep3B cell xenograft model.

Tumor-vessel normalization reportedly improves chemotherapeutic drug delivery in mouse models, decreases tumor growth, and increases survival [48,49,50]. Our present study shows that the vinorelbine or everolimus induces tumor-vessel normalization, and the addition of the everolimus (or sirolimus) to the vinorelbine significantly improves the antitumor activity of the single agents and the OS of the mice bearing orthotopic HCC tumors. Our mechanistic studies have demonstrated that compared with the antitumor activity of the everolimus alone, the marked antitumor activity of the combination was not affected by the further suppression of the mTOR-signaling pathway. These findings are strongly supported by a previous report showing that the combination of temsirolimus with vinblastine potently inhibits the growth of HCC tumor xenografts by decreasing the expressions of p70S6K and survivin [43]. In addition, the combination of sirolimus and vinblastine [51] and a liposomal formulation encapsulating the everolimus and vinorelbine [52] were effective against the growth of human neuroblastoma and RCC, respectively. Our study and others hold particular significance, especially when considering the weak monotherapeutic activity of mTOR inhibitors (everolimus and sirolimus), which has already been demonstrated in HCC [36,37,38], alongside the efficacy of the vinorelbine in other cancer types [27,28,29,30,46,52]. Further investigation is needed to better understand the interactions between the everolimus (or sirolimus) and vinorelbine in the tumor response and the mechanisms underlying their antitumor activity and blood-vessel normalization. The synergistic growth inhibition of everolimus/vinorelbine may be explained, at least in part, by blood-vessel normalization, which aids in the delivery of the drugs to the tumor cells, resulting in the downregulations of the mTOR-signaling pathway, survivin, and cell-cycle regulators, as well as an increase in apoptosis.

To reduce the toxicity associated with combination therapy, the dosage of each individual drug often needs to be reduced in the combination regimen [53]. Using HCC PDX models and treatment scenarios, we provide evidence of an enhanced tumor response with mild hepatic toxicity when the vinorelbine is added to the everolimus. In in vivo studies, everolimus is typically administered daily via the oral route at 1–5 mg/kg/day [54]. Similarly, the usual dose of vinorelbine is 4.8–5 mg/kg/week via intravenous or intraperitoneal routes [55,56]. In the clinic, the most common side effects of everolimus include nausea, anorexia, diarrhea, stomatitis, pneumonitis, and a rash [17,57]. The common side effects of a standard dose of vinorelbine include neutropenia, peripheral neuropathy, and gastrointestinal (GI) complications [33,58]. A previous attempt to reduce the side effects associated with everolimus/vinorelbine has led to the development of a tumor-targeted liposomal formulation with a nominal dose of everolimus (1 mg/kg; 3×/week) and a low dose of vinorelbine (0.475 mg/kg; 3×/week) [52]. This liposomal formulation encapsulating the everolimus and vinorelbine induced remarkable tumor inhibition in RCC tumor xenografts with minimal side effects [52]. This liposomal formation could be a promising approach for the treatment of HCC, where there is a clear unmet need for effective therapies.

The everolimus and sirolimus exert potent immunosuppressant effects and are commonly used as immunosuppressive drugs [57]. They inhibit the mTOR pathway, which plays a crucial role in the cell growth, proliferation, angiogenesis, survival of malignant tumors, and the balance between effector T cells and regulatory T cells (Tregs). In patients with metastatic renal cell carcinoma, the everolimus treatment caused immunological alterations in circulating immune cells, including increases in the numbers of Tregs and monocytic myeloid-derived suppressor cells and decreases in the numbers of immunoregulatory natural killer cells, classical CD141+ (cDC1), and CD1c+ (cDC2) dendritic cell subsets [59]. The immunosuppressive effects of the everolimus and sirolimus limit their use in cancer treatment. Future studies are needed to determine whether a low dose of everolimus or sirolimus in combination with a metronomic dose of vinorelbine or a liposomal formulation encapsulating everolimus and vinorelbine [52] can help mitigate immune suppression and boost T-cell immunity without compromising their antitumor efficacies.

In summary, using clinically relevant PDX HCC models and treatment scenarios, our study showed that the combination of mTOR inhibitors (everolimus and sirolimus) with vinorelbine induces a remarkable tumor response. Notably, p70S6K/4EBP1 and survivin are widely overexpressed in HCC (~70% of cases), and these factors are significantly associated with the pathological grade of HCC (e.g., high expression of survivin is an independent indicator of a poor prognosis in HCC patients) [37,60]; thus, this highly effective combination of everolimus/vinorelbine may prove to be particularly relevant in clinical settings of HCC as a rational treatment option. The clinical significance of this study is further amplified by the cost-effectiveness for utilizing two established, inexpensive, and simple-to-deliver drugs (vinorelbine and, especially, sirolimus), making treatment accessible and affordable for patients. Taking these results together, this study provides a strong rationale for future phase I/II clinical trials aimed at improving the efficacy of frontline therapy for HCC patients.

## 4. Materials and Methods

The reagents, HCC cell isolations and cultures, vessel perfusion studies, tumor harvesting and processing, immunohistochemistry (IHC), slide imaging and quantification, western blot analysis, orthotopic models, and statistical analyses were prepared or performed as previously described [27,34,61,62].

### 4.1. Reagents

Vinorelbine (Navelbine^®^) (10 mg/mL) was obtained from Pierre Fabre Medicament (Boulogne, France) and was dissolved in PBS to a final concentration of 0.375 mg/mL before use. Everolimus (RAD001) was obtained from Novartis (Basel, Switzerland) and dissolved in a vehicle (30% Captisol^®^ in water) to achieve the appropriate concentration for the treatments.

Antibodies against AKT (#9272), p70S6K (#9202), survivin (#2803), S6R (#2217), Rb (#9313), Cyclin B1 (#4138), eIF4E (#9742), Cdc25C (#4688), cleaved caspase 3 (#9661), cleaved caspase 7 (#9491), cleaved PARP (#5625), Cyclin D1 (#2978), Cdc2 (#9112), and α-Tubulin (#2144) and phosphorylation-specific antibodies against RB Ser807/811 (#9308), AKT Ser473 (#9271), mTOR Ser2448 (#5536), p70S6K Thr421/424 (#9204), S6R Ser235/236 (#4858), 4EBP1 Thr70 (#9455), Histone H3 Ser10 (#9701), Cyclin D1 Thr286 (#3300), Cdc25C Ser216 (#4901), Cdc2 Tyr15 (#9111), eIF4E Ser209 (#9741), and ERK1/2 Thr202/Tyr204 (#4370) were obtained from Cell Signaling Technology (Beverly, MA, USA). RKIP (#37-2100) was obtained from Invitrogen. p-Cdk2 Thr14/Tyr15 (sc-28435-R), ERK1/2 (sc-94), and p27 (sc-528) were obtained from Santa Cruz Biotechnology Inc. (Santa Cruz, CA, USA). Anti-mouse CD31 (#2502) antibody was purchased from BioLegend (San Diego, CA, USA).

### 4.2. Patient-Derived Xenograft (PDX) HCC Models

This study received ethics board approval from the SingHealth Centralised Institutional Review Board (ethics code: CIRB #2006/435/B; approval date: 2 October 2018). All the mice were maintained in accordance with the guidelines outlined in the *Guide for the Care and Use of Laboratory Animals*, published by the National Institutes of Health, USA [63].

HCC tumors have previously been used to create xenograft lines [61]. A total of 15 HCC xenograft lines (HCC01–0708, HCC05–0411B, HCC05–0614, HCC06–1009, HCC09–0913, HCC13–0109, HCC13–0212, HCC19–0509, HCC19–0913, HCC24–0309, HCC25–0705A, HCC26–0808B, HCC27–1014, HCC29–0909A, and HCC30–0805B) were selected to establish tumors in male SCID mice (InVivos, Singapore) aged from 9 to 10 weeks. Briefly, under sterile conditions, the HCC xenograft tumors were minced into fine fragments that would pass through an 18-gauge needle and then mixed at a ratio of 1:1 (*v*/*v*) with Matrigel^®^ (Corning Inc., Corning, NY, USA) to result in a total volume of 150 µL per injection. The tissue mixture was subcutaneously injected in both flanks of each mouse. The growth of the xenograft tumors was monitored at least twice weekly until the tumor sizes reached approximately 170–200 mm^3^.

The mice were housed in negative-pressure isolators set at 23 °C and 43% humidity, with 12 h light/dark cycles and were provided with sterilized food and water ad libitum. All the studies were performed in accordance with IACUC-approved procedures.

### 4.3. Drug Treatment and Efficacy of Everolimus/Vinorelbine in Ectopic PDX HCC Models

To assess the time-dependent effects of the vinorelbine on the apoptosis, cell proliferation, tumor blood-vessel density, and blood-vessel normalization, mice bearing HCC13–0212 tumors were treated intraperitoneally with the vehicle (PBS) or 3 mg/kg of vinorelbine. The treatment started when the tumor sizes reached approximately 250–300 mm^3^. The tumors were harvested at 0, 24, 36, 48, 60, and 72 h after treatment for IHC.

For the combination therapy, mice bearing the indicated HCC xenograft lines were randomized into four treatment groups (*n* = 8–10) and treated as follows: (a) vehicle plus PBS, (b) a standard dose of 2 mg/kg of everolimus plus PBS, (c) a dose of 3 mg/kg of vinorelbine plus the vehicle, or (d) 2 mg/kg of everolimus plus 3 mg/kg of vinorelbine (everolimus/vinorelbine) for the indicated time. The treatment regimens included the daily oral administration of the vehicle and everolimus, while the PBS and vinorelbine were administered intraperitoneally twice per week (once every 3.5 days). All the treatments started when the tumor sizes reached approximately 170–200 mm^3^. The tumor growth, bodyweight, and signs of illness were monitored and recorded, as described in previous studies [27,28,29,30]. At the end of the experiment, the mice were sacrificed, and the tumors were resected, weighed, and recorded. The harvested tumors were divided into two parts: One part was snap-frozen in liquid nitrogen for molecular analyses, and the other part was fixed in 10% formalin and processed for IHC. The experiments were repeated on specific HCC xenografts by replacing the everolimus with the sirolimus.

To determine the efficacies of the everolimus, vinorelbine, and everolimus/vinorelbine treatments in the HCC PDX models, the T/C ratio was calculated by dividing the median weight of the drug-treated tumors (T) by that of the vehicle-treated tumors (C) at the end of the treatment. T/C ratios of <0.42 were considered as being active, in accordance with the criteria established by the Cancer Therapy Evaluation Program (CTEP) of the Investigational Drug Branch (IDB) at the National Cancer Institute [30].

### 4.4. Vessel Perfusion and Hypoxia Studies

Vessel perfusion and hypoxia studies were performed according to the protocol previously described in [27]. Mice bearing tumors (vehicle- or drug-treated) were intravenously injected with 100 mg of biotinylated tomato lectin (derived from *Lycopersicon esculentum*) (Vector Labs, Newark, CA, USA, #B-1175) prepared in 100 μL of 0.9% NaCl, followed by intraperitoneal injection with 60 mg/kg of pimonidazole hydrochloride. The tumors were harvested 2 h after the lectin perfusion and hypoxia injection, fixed in 10% formalin, and processed for subsequent experiments. To visualize the productive microvessels, IHC was performed using the streptavidin–biotin peroxidase complex method according to the manufacturer’s instructions (Lab Vision Corporation, Fremont, CA, USA). To determine the extent of the hypoxia, the hypoxic regions within the tumors were identified by staining the sections with the Hypoxyprobe Plus Kit HP2 according to the manufacturer’s instructions (Hypoxyprobe Inc., Burlington, MA, USA). For the quantification analysis, 10 random 0.159 mm^2^ fields were captured at a magnification of 100× and counted on each IHC-stained slide.

### 4.5. Immunohistochemistry (IHC)

IHC was performed according to a previously described protocol [62]. The slides were stained with antibodies against CD31 (Cell Signaling Technology, #77699); p-histone H3 Ser10 (Cell Signaling Technology, #9701); and cleaved PARP (Cell Signaling Technology, #5625) to assess the microvessel density, cell proliferation, and apoptosis, respectively. At least 10 fields were randomly captured at a magnification of 100× on each IHC-stained slide using an Olympus BX60 microscope (Olympus, Tokyo, Japan). To quantify the mean of the microvessel density, the p-histone H3 Ser10, and the cleaved PARP cells, all the positively stained cells in the captured images were counted and expressed as a percentage value compared with the total number of cells in that region.

### 4.6. Cell Cultures

Single cells were isolated from HCC13–0109 and HCC25–0705A tumors and cultured as monolayer cultures in high-glucose Dulbecco’s modified Eagle medium (DMEM) supplemented with 10% fetal bovine serum (FBS) and 1% penicillin–streptomycin at 37 °C with 5% CO_2_, as previously described [62].

### 4.7. Propidium Iodide (PI) Flow Cytometry Analysis

HCC13–0109 and HCC25–0705A cells were prepared as previously described [62]. They were plated at a density of 5 × 10^5^ and then treated with the vehicle, 0.1 µM everolimus, or 1 nM vinorelbine in growth medium for 24 h. The cells were fixed in 70% ethanol at 4 °C for 24 h. Subsequently, the cells were rehydrated, washed in PBS, and treated with ribonuclease A (RNaseA; 1 mg/mL), followed by staining with propidium iodide (PI) (100 μg/mL). The fluorescence intensities of the stained cells were measured using a FACSCalibur™ flow cytometer (BD, San Jose, CA, USA). The data were analyzed using BD CellQuest™ Pro software (Version 4.1; BD, San Jose, CA, USA), as described previously [27]. For every measurement, 10,000 events were collected, and the gating was set to exclude the cell doublets. The DNA contents of certain phases are shown as percentages compared with the total DNA content within the gate. The average of three separate experiments is presented.

### 4.8. Serum Analysis

Sera were derived from the mice treated with the vehicle, everolimus, vinorelbine, and everolimus/vinorelbine for 14 days. The sera were collected to determine the levels of total bilirubin (TBIL), alkaline phosphatase (ALP), alanine aminotransferase (ALT), aspartate aminotransferase (AST), albumin (ALB), creatinine (Cre), glucose (GLU), and blood urea nitrogen (BUN) using the preventive care profile plus (Abaxis Inc., Union City, CA, USA) according to the manufacturer’s instructions.

### 4.9. Western Blot Analysis

To assess the changes in the protein expressions between the vehicle- and drug-treated tumors, frozen tissues were homogenized in a buffer containing 50 mM Tris-HCl (pH 7.4), 150 mM NaCl, 0.5% NP-40, 1 mM EDTA, and 25 mM NaF supplemented with protease inhibitors and 10 mM Na_3_VO_4_. Eighty micrograms of protein per sample were resolved using SDS–PAGE and transferred to a nitrocellulose membrane, as described in [27]. The blots were incubated with the indicated primary antibodies and horseradish-peroxidase-conjugated secondary antibodies. The blots were then visualized with a chemiluminescent detection system (Amersham Pharmacia Biotech, Amersham, UK) and exposed to autoradiography film.

### 4.10. Efficacies of Everolimus, Vinorelbine, and Everolimus/Vinorelbine in Orthotopic PDX HCC Models

The HCC13–0212, HCC24–0309, and HCC25–0705A orthotopic models were generated as previously described [34]. Briefly, SCID mice were anesthetized with a ketamine/diazepam solution (50 mg/kg of ketamine hydrochloride; Rotexmedica, Trittau, Germany; and 5 mg/kg of diazepam (Atlantic), I.M.). Baytril^®^ at 5 mg/kg was given intramuscularly. Under sterile conditions, a small upper midline laparotomy was performed to exteriorize the left lobe of the liver. Approximately 5 × 10^6^ tumor cells (in 30 µL of a medium–Matrigel mixture) were implanted in the lobe of the liver, using a 27-gauge needle. The incision was closed using a running suture of 5-0 silk. For the survival study, mice bearing HCC tumors (*n* = 10/group) were treated with the vehicle, everolimus, vinorelbine, or everolimus/vinorelbine according to the treatment conditions described above. The treatments started when the tumor sizes were approximately 100–150 mm^3^. The bodyweight and OS were monitored daily. The mice were sacrificed when they became moribund.

### 4.11. Statistical Analysis

The differences between the tumor volumes, tumor weights, and bodyweights at sacrifice and the means of the p-histone H3 Ser10-, cleaved PARP-, CD31-, and lectin-positive cells were compared. Student’s *t*-test was used for comparisons between two groups. One-way analysis of variance (ANOVA) followed by the Tukey–Kramer post hoc test were used when comparing more than two groups. The error bars are given based on the calculated SD values. For the survival analysis, the log-rank test was used. A *p*-value of <0.05 was considered as being statistically significant.

## Figures and Tables

**Figure 1 ijms-25-00017-f001:**
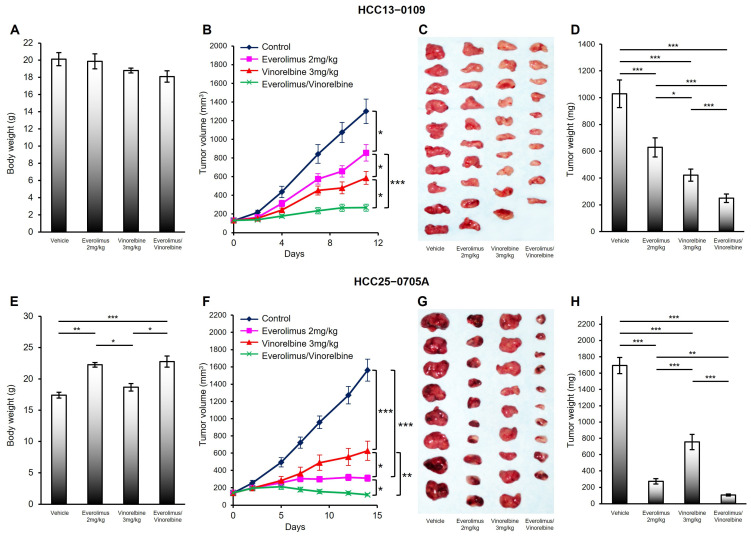
Effects of everolimus, vinorelbine, and everolimus/vinorelbine on tumor growth in the HCC13–0109 and HCC25–0705A PDX models. HCC tumors were subcutaneously implanted in SCID mice, as described in Section 4. Mice bearing the indicated tumors were randomly divided into four groups and orally treated with 200 μL of the vehicle, 2 mg/kg of everolimus once daily, 3 mg/kg of vinorelbine once every 3.5 days, or 2 mg/kg of everolimus plus 3 mg/kg of vinorelbine on the indicated days. Each treatment arm comprised 8–10 independent tumor-bearing mice. (**A**,**E**) The mean bodyweights ± SEs at sacrifice; (**B**,**F**) the mean tumor volumes ± SEs at given time points; (**C**,**G**) representative vehicle-, everolimus-, vinorelbine-, and everolimus/vinorelbine-treated tumors; and (**D**,**H**) the mean corresponding tumor weights ± SEs are shown. Different asterisks (*) indicate significant differences (* *p* < 0.05; ** *p* < 0.01; *** *p* < 0.001; ANOVA followed by Tukey’s test).

**Figure 2 ijms-25-00017-f002:**
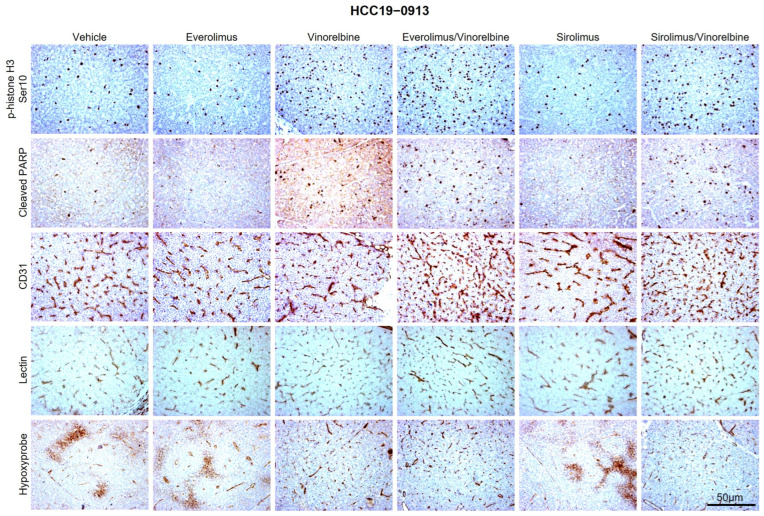
Effects of everolimus (or sirolimus), vinorelbine, and everolimus (or sirolimus)/vinorelbine on cell proliferation, apoptosis, blood-vessel density, blood-vessel normalization, and tumor hypoxia. HCC19–0913 tumors were subcutaneously implanted in SCID mice, as described in Section 4. Mice bearing the indicated tumors were randomly divided into four groups and orally treated with 200 μL of the vehicle, 2 mg/kg of everolimus (or sirolimus) once daily, 3 mg/kg of vinorelbine once every 3.5 days, or 2 mg/kg of everolimus (or sirolimus) plus 3 mg/kg of vinorelbine for the indicated days. Each treatment arm comprised 8–10 independent tumor-bearing mice. Tumors collected 2 h after the last treatments were processed for IHC, as described in Section 4. Representative images of tumor sections from vehicle- and drug-treated mice stained for p-histone H3 Ser10, cleaved PARP, CD31 (blood vessels), lectin, and Hypoxyprobe antibodies are shown. Images were captured using an Olympus BX60 microscope (Olympus, Tokyo, Japan). Scale bars: 50 μM.

**Figure 3 ijms-25-00017-f003:**
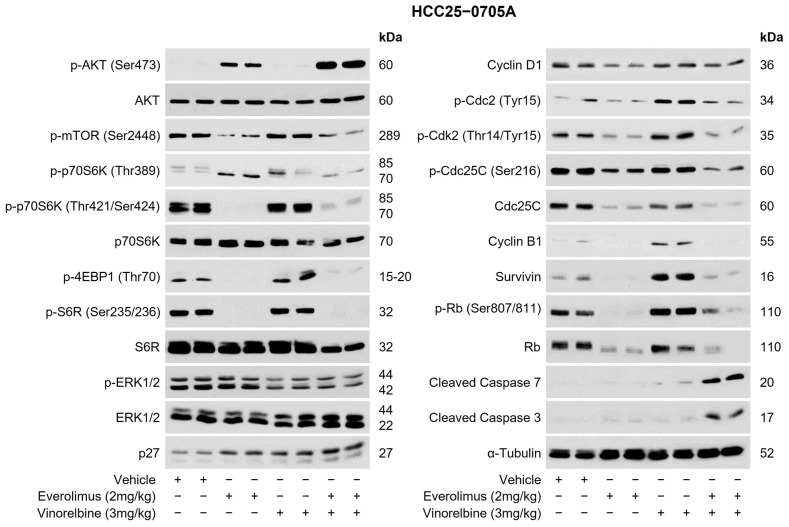
Effects of everolimus, vinorelbine, and everolimus/vinorelbine on AKT/p70S6K/4EBP1, ERK1/2, and cell-cycle regulators. HCC25–0705A tumors were subcutaneously implanted in SCID mice, as described in Section 4. Mice bearing HCC25–0705A tumors were randomly divided into four groups and orally treated with 200 µL of the vehicle, 2 mg/kg of everolimus once daily, 3 mg/kg of vinorelbine once every 3.5 days, or 2 mg/kg of everolimus plus 3 mg/kg of vinorelbine for the indicated days. Each treatment arm comprised 8–10 independent tumor-bearing mice. Tumors collected 2 h after the last treatments were subjected to a western blot analysis, as described in Section 4. Representative blots incubated with the indicated antibodies and molecular markers of the proteins (kDa) are shown.

**Figure 4 ijms-25-00017-f004:**
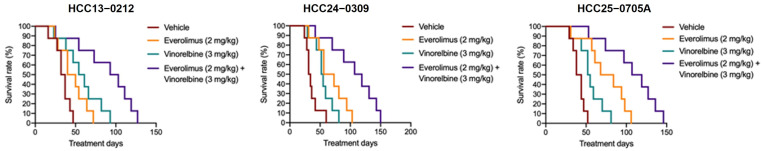
Effects of everolimus, vinorelbine, and everolimus/vinorelbine on the survival of HCC13–0212, HCC24–0309, and HCC25–0705A orthotopic models. HCC orthotopic models were established as previously described in [34]. These models were treated with everolimus, vinorelbine, or everolimus/vinorelbine for the indicated days, as described in Section 4. Each treatment group consisted of 10 mice. Treatments commenced when the tumor sizes reached approximately 100–150 mm^3^. Kaplan–Meier survival curves are presented. Everolimus/vinorelbine significantly improved the OS of the mice (*p* < 0.01; log-rank test).

**Table 1 ijms-25-00017-t001:** Effects of everolimus (or sirolimus), vinorelbine, and everolimus (or sirolimus)/vinorelbine on the tumor burden in the HCC PDX models. The indicated tumors were subcutaneously implanted in SCID mice, as described in Section 4. Mice bearing the indicated tumors were randomly divided into four groups and orally treated with 200 μL of the vehicle, 2 mg/kg of everolimus (or sirolimus) once daily, 3 mg/kg of vinorelbine once every 3.5 days, or 2 mg/kg of everolimus (or sirolimus) plus 3 mg/kg of vinorelbine for the indicated days. Each treatment arm comprised 8–10 independent tumor-bearing mice. The mean tumor weight was recorded, and the efficacy of each treatment was determined using the T/C ratio, where T and C represent the median tumor weights of the drug-treated and vehicle-treated mice, respectively, at the end of the treatment. The T/C ratios are shown.

HCC PDX Model	Treatments and T/C Ratios					
	Control	Everolimus 2 mg/kg QD	Vinorelbine 3 mg/kg Q3.5D	Everolimus/ Vinorelbine	Sirolimus 2 mg/kg QD	Sirolimus/ Vinorelbine
HCC05–0411B	1	0.1883	0.6707	0.1389		
HCC06–1009	1	0.2931	0.5650	0.2031	0.3958	0.1919
HCC24–0309	1	0.1218	0.5142	0.0698		
HCC26–0808B	1	0.1886	0.5775	0.1780		
HCC13–0212	1	0.3359	0.1966	0.0656	0.3785	0.1292
HCC19–0509	1	0.3754	0.3348	0.2319		
HCC25–0705A	1	0.1622	0.3869	0.0635		
HCC01–0708	1	0.5767	0.7542	0.3345		
HCC05–0614	1	0.5395	0.5105	0.1523		
HCC09–0913	1	0.5645	0.6651	0.3314		
HCC13–0109	1	0.6108	0.3092	0.2419		
HCC19–0913	1	0.5770	0.3192	0.1669	0.5888	0.1691
HCC27–1014	1	0.7707	0.5866	0.3332		
HCC29–0909A	1	0.6287	0.5671	0.3680		
HCC30–0805B	1		0.6080		0.6176	0.2041

**Table 2 ijms-25-00017-t002:** Effects of everolimus, vinorelbine, and everolimus/vinorelbine on liver- and kidney-injury-related parameters. Sera derived from mice bearing HCC13–0109 treated with the vehicle, everolimus, vinorelbine, and everolimus/vinorelbine for 14 days were analyzed using preventive care profile plus (Abaxis Inc., Union City, CA, USA) according to the manufacturer’s instructions. The levels of total bilirubin (TBIL), alkaline phosphatase (ALP), alanine aminotransferase (ALT), aspartate aminotransferase (AST), and albumin (ALB) served as markers of the overall liver function. The levels of creatinine (Cre), glucose (GLU), and blood urea nitrogen (BUN) served as indicators of kidney function.

Serum Marker	Unit	Control	Everolimus 2 mg/kg QD	Vinorelbine 3 mg/kg Q3.5D	Everolimus/ Vinorelbine
BUN	(mg/dL)	13.8	13.1	16.5	16.8
CRE	(mg/dL)	0.45	0.38	0.49	0.46
ALT	(U/L)	38.7	54.5	69.5	62.8
ALP	(U/L)	54.1	73.7	88.2	89.2
AST	(U/L)	195	254.6	279.0	308.5
TBIL	(mg/dL)	0.3	0.32	0.38	0.39
GLU	(mg/dL)	163.2	170.4	156.9	188
ALB	(g/dL)	4.0	3.75	3.7	3.6

**Table 3 ijms-25-00017-t003:** Cell-cycle analysis of HCC13–0109 and HCC25–0705A cells. Single cells isolated from HCC13–0109 and HCC25–0705A tumors were plated at a density of 5 × 10^5^ and then treated with vehicle, 0.1 µM everolimus, or 1 nM vinorelbine in a growth medium for 24 h. The cells were fixed in 70% ethanol for 24 h, rehydrated, washed in PBS, and treated with ribonuclease A, followed by staining with PI. The cell-cycle analysis was performed as described in [27]. The DNA contents of the specific phases are shown as percentages compared with the total DNA content within the gate. The averages of three separate experiments are presented.

HCC Cells	Drugs	Stage of Cell Cycle			
		Sub G1	G1	S	G2/M
HCC13–0109	Vehicle	1.56 ± 0.31	63.70 ± 3.50	3.80 ± 0.42	30.94 ± 2.10
	Everolimus 0.1 μM	1.69 ± 0.35	74.52 ± 4.21	1.84 ± 0.28	21.95 ± 1.86
	Vinorelbine 1 nM	10.53 ± 1.14	42.59 ± 3.65	2.20 ± 0.54	44.68 ± 5.11
	Everolimus 0.1 μM + Vinorelbine 1 nM	17.68 ± 1.26	32.40 ± 1.89	1.54 ± 0.28	48.38 ± 4.23
HCC25–0705A	Vehicle	1.91 ± 0.30	57.71 ± 3.68	11.90 ± 1.34	28.48 ± 1.89
	Everolimus 0.1 μM	12.06 ± 1.06	66.53 ± 4.29	3.0 ± 0.23	18.41 ± 1.22
	Vinorelbine 1 nM	13.63 ± 1.88	43.98 ± 2.57	5.16 ± 0.34	37.23 ± 2.80
	Everolimus 0.1 μM + Vinorelbine 1 nM	14.59 ± 2.43	41.35 ± 2.64	5.94 ± 0.21	38.12 ± 2.53

## Data Availability

The datasets used and analyzed in the current study are available within the manuscript and its Appendix A.

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
