# Peer review of "Everolimus Acts in Synergy with Vinorelbine to Suppress the Growth of Hepatocellular Carcinoma"

_ijms, 2023, doi:10.3390/ijms25010017_

Round 1

Reviewer 1 Report

Comments and Suggestions for Authors

Dear authors, I have carefully reviewed your manuscript entitled “Everolimus Acts in Synergy with Vinorelbine to Suppress the Growth of Hepatocellular Carcinoma”.

The manuscript is both informative and well-written, offering a comprehensive exploration of the synergistic effects of Everolimus and Vinorelbine in suppressing HCC growth. The experiments described in the study are well-introduced, logically designed, and the conclusions drawn by the authors are soundly supported by the included data.

I have provided point-by-point comments below for the authors' consideration and response. I believe that once addressed, these comments will further enhance the overall quality of the manuscript.

Point-by-point comments:

Introduction line 29. While it is true that Sorafenib has been considered the standard of care for patients with advanced unresectable HCC, its efficacy is limited, as the authors mentioned in line 48. To provide the reader with a stronger message I would suggest to combine the text at line 29 and line 48.

Line 97 (and line 358). The authors do not provide enough information to evaluate if 3mg/kg of vinorelbine qualify as a metronomic therapy, either include this information or remove.

Line 100, The authors should state in the text, as well as in the figure legend of Table 1, that these are PDX models. As it stands it gives the idea that different models (PDX plus something else) was used in the study.

Line 329. I found the statement “Our data show that everolimus/vinorelbine is superior to everolimus or vinorelbine in improving the OS of mice with HCC tumors” misleading as the authors analyzed xerograph mice and this should be included in the text, please change.

Line 361. Please add reference supporting the statement: “Drug Evaluation Branch of the Division of Cancer Treatment, National Cancer Institute”.

Line 466. Please add reference supporting the statement “has already been demonstrated in HCC, alongside the efficacy of vinorelbine in other cancer types”.

Line 491. I am not sure that the data presented by the authors fully support the statement: “vinorelbine induces a remarkable tumor response without additional toxicity”. Specifically, on the “toxicity”. The data shown by the authors are impressive, however no data on toxicity were shown. To support this statement the authors should include analysis of liver enzymes from the sera/plasma prepared from the xerograph (ALT, AST, GLDH, Urea and bilirubin) and include them in the manuscript.

Minor remark:

Line 83. The full stop was used twice

Reviewer 2 Report

Comments and Suggestions for Authors

In this manuscript, the authors reported that everolimus acts in synergy with vinorelbine to suppress the growth of hepatocellular carcinoma. The reviewer’s concerns are as follows:

1. The authors claimed that vinorelbine arrests cells at the mitotic phase, induces apoptosis, and normalizes tumor blood vessels but upregulates survivin and activates the mTOR/p70S6K/4EBP1 pathway. However, they only analyzed the expression of related markers by IHC analysis and made these conclusions. The data are not enough to support the conclusion. Cell cycle analysis (e.g., flow cytometry) and apoptosis assays (e.g., PI/ANNEXIN V staining) should performed.

2. In Figure 1, the images of xenograft should be presented.

3. Whether did these drug treatments change the body weight of mice?

4. Some supplemental data should be presented in the manuscript. For example, the analysis of tumor weight.

5. mTOR pathway is associated with the sorafenib resistance of HCC (PMID: 35977595, PMID: 35192933). Thus, it’s possible that the inhibitors used in this study enhance the sorafenib effect. It’s suggested to discuss.

6. It’s suggested to add molecular markers in all western blot.

7. It’s suggested to check the grammar and correct the typo errors.

8. The supplemental data are too many. It’s suggested to re-organize the data.

Comments on the Quality of English Language

It’s suggested to check the grammar and correct the typo errors.

Reviewer 3 Report

Comments and Suggestions for Authors

The manuscript entitled " Everolimus Acts in Synergy with Vinorelbine to Suppress the Growth of Hepatocellular Carcinoma " has been reviewed.

It is an interesting paper. It is also logical and well-written, although some parts are not strongly validated.

Comments on the Quality of English Language

none

Reviewer 4 Report

Comments and Suggestions for Authors

Line 51. can you provide reference for early metastatic spread of HCC.

Line 344-355. can you explain how did you chose HCC in your study which are refractory to Sorafenib or lenvatinib.

Did the authors note any relation of the drugs on liver function tests?

Line 447-450. What authors feel the reasons for the differences in results between their study and study done by Qian Zhou et al (reference 38).

Round 2

Reviewer 2 Report

Comments and Suggestions for Authors

The authors have addressed my concerns.